# Consecutive Prostate Cancer Specimens Revealed Increased Aldo–Keto Reductase Family 1 Member C3 Expression with Progression to Castration-Resistant Prostate Cancer

**DOI:** 10.3390/jcm8050601

**Published:** 2019-05-01

**Authors:** Yu Miyazaki, Yuki Teramoto, Shinsuke Shibuya, Takayuki Goto, Kosuke Okasho, Kei Mizuno, Masayuki Uegaki, Takeshi Yoshikawa, Shusuke Akamatsu, Takashi Kobayashi, Osamu Ogawa, Takahiro Inoue

**Affiliations:** 1Department of Urology, Kyoto University Graduate School of Medicine, Kyoto 606-8507, Japan; urozaki@kuhp.kyoto-u.ac.jp (Y.M.); goto@kuhp.kyoto-u.ac.jp (T.G.); k_okasho@kuhp.kyoto-u.ac.jp (K.O.); km1207@kuhp.kyoto-u.ac.jp (K.M.); uegaki57@kuhp.kyoto-u.ac.jp (M.U.); urotake9@kuhp.kyoto-u.ac.jp (T.Y.); akamats@kuhp.kyoto-u.ac.jp (S.A.); selecao@kuhp.kyoto-u.ac.jp (T.K.); ogawao@kuhp.kyoto-u.ac.jp (O.O.); 2Department of Diagnostic Pathology, Kyoto University Hospital, Kyoto 606-8507, Japan; tera1980@kuhp.kyoto-u.ac.jp (Y.T.); sshibuya@kuhp.kyoto-u.ac.jp (S.S.)

**Keywords:** AKR1C3, hormone-naïve prostate cancer, castration-resistant prostate cancer, immunohistochemistry, tissue microarray

## Abstract

Aldo-keto reductase family 1 member C3 (AKR1C3) is an enzyme in the steroidogenesis pathway, especially in formation of testosterone and dihydrotestosterone, and is believed to have a key role in promoting prostate cancer (PCa) progression, particularly in castration-resistant prostate cancer (CRPC). This study aims to compare the expression level of AKR1C3 between benign prostatic epithelium and cancer cells, and among hormone-naïve prostate cancer (HNPC) and CRPC from the same patients, to understand the role of AKR1C3 in PCa progression. Correlation of AKR1C3 immunohistochemical expression between benign and cancerous epithelia in 134 patient specimens was analyzed. Additionally, correlation between AKR1C3 expression and prostate-specific antigen (PSA) progression-free survival (PFS) after radical prostatectomy was analyzed. Furthermore, we evaluated the consecutive prostate samples derived from 11 patients both in the hormone-naïve and castration-resistant states. AKR1C3 immunostaining of cancer epithelium was significantly stronger than that of the benign epithelia in patients with localized HNPC (*p* < 0.0001). High AKR1C3 expression was an independent factor of poor PSA PFS (*p* = 0.032). Moreover, AKR1C3 immunostaining was significantly stronger in CRPC tissues than in HNPC tissues in the same patients (*p* = 0.0234). Our findings demonstrate that AKR1C3 is crucial in PCa progression.

## 1. Introduction

Prostate cancer (PCa) is one of the most commonly diagnosed malignancies and the second leading cause of cancer deaths in the United States [1]. In Japan, the mortality rate of PCa is the sixth among those of all male malignancies, although the estimated incidence rates have slightly declined, possibly due to reduced prostate-specific antigen (PSA) screening, the same as in the United States [2]. In the early 1940s, Huggins and Hodges demonstrated growth and survival of PCa to depend on androgens [3]. Therefore, androgen deprivation therapy (ADT) has been a standard clinical procedure for the control of PCa growth, with patients mostly responsive to ADT at the beginning of the therapy, which is also called hormone-naïve prostate cancer (HNPC); however, most of those patients relapse thereafter, developing castration-resistant prostate cancer (CRPC). ADT is required to treat advanced PCa and biochemical recurrent cases after curative radical treatment; however, despite the reduction of serum testosterone (T) to castration levels and an observed tumor response in 80%–90% of the patients, residual concentrations of intratumoral 5α-dihydrotestosterone (DHT) remain at 10%–40% of the pre-ADT levels in castration-resistant and hormone-naïve states [4,5]. The amount of residual androgens is substantial for triggering androgen receptor (AR) signaling, AR target gene expression, and cancer cell proliferation [6]. The de-novo pathway, which commences with cholesterol requiring multiple androgen synthetic enzymes, may be a result of the intratumoral androgen function; however, whether the complete repertoire of synthetic enzymes is required to generate androgens from cholesterol remains to be fully elucidated [7,8]. Circulating adrenal androgens, which are abundant in the form of dehydroepiandrosterone (DHEA) and with sulfated modification, dehydroepiandrosterone sulfate (DHEA-S), are other significant points of origin of the androgens. The acquired capacity of converting the adrenal androgens to more potent forms is a characteristic of CRPC. Therefore, abiraterone acetate, which is a potent CYP17A1 (17-hydroxylase/17, 20-lyase) inhibitor, is effective against CRPC, and has been recently approved for treating metastatic HNPC. Type 5 17α-hydroxysteroid dehydrogenase, in another name, aldo-keto reductase family 1 member C3 (AKR1C3) is a crucial enzyme in the steroidogenesis pathway. It catalyzes Δ^4^-androsetene-3, 17-dione to T, DHT to 5α-androstane-3α, 17β-diol (3α-diol), and 3α-diol to androsterone; thus, it plays an important role in the formation of T and DHT [9]. Additionally, AKR1C3 can also reduce the weak estrogen, estrone, to the potent estrogen, 17β-estradiol, which might induce local estrogen production, contributing to PCa occurrence [10,11]. Estrogen and estrogen receptor (ER) (ER alpha: ERα and ER beta: ERβ) axes play an important role in both prostate carcinogenesis and progression to CRPC [12,13]. Although PCa co-expresses classical ERs, ERα and ERβ, and also non-genomic receptor, GRP30, complex interactions between ERs and AR, and those among various ligands in PCa cells need further investigation [12,13]. Moreover, AKR1C3 is known as prostaglandin (PG) F synthase that catalyzes the conversion of PGD_2_ to 11-βPGF2α and PGF2α prostanoids, hence contributing to proliferation and radio-resistance in PCa cells [14,15]. All these issues imply that AKR1C3 could have a potential role in PCa biology. Several studies have demonstrated that AKR1C3 expression levels are elevated in PCa cells than in benign cells; moreover, it is highly expressed in the CRPC cell lines and human CRPC tissues rather than in the hormone-naïve ones [9,16,17,18,19,20,21,22]. Nevertheless, most reports have focused on the PCa tissues derived from different patients and compared the expression levels in normal/benign prostate hyperplasia tissues, localized cancer, and metastatic CRPC. This study aimed to compare the expression level of AKR1C3 between normal prostatic epithelium and cancer cells in the same patients. Moreover, we evaluated AKR1C3 expression and PSA progression-free survival after radical prostatectomy. We also investigated the expression level in hormone-naïve cancer and advanced CRPC in the same patients, to better understand the role of AKR1C3 in PCa progression.

## 2. Materials and Methods

### 2.1. Human Prostate Tissue Samples

All PCa patients included in this study were Japanese patients. Tissue-microarrays (TMAs) consisted of 175 radical prostatectomy (RP) specimens of patients with hormone-naïve PCa, who received RP between December 2004 and October 2012 at Kyoto University Hospital [23]. The TMA was developed with one core from each case. This study included 134 cases, which had both cancer and non-cancer tissues in each TMA core. We defined PSA failure as two consecutive measurements of PSA levels of ≥0.2 ng/mL, and the date of PSA failure as the time of the first measurement of PSA level of ≥0.2 ng/mL. When PSA levels after surgery did not decline below 0.2 ng/mL, we defined the date of PSA failure by the time of surgery. Consecutive prostate samples derived from 11 patients, both at hormone-naïve and castration-resistant states, were evaluated. HNPC specimens consisted of samples from needle biopsy or from transurethral resection of the prostate (TUR-P). CRPC specimens were collected from the TUR-P samples against urinary retention or gross hematuria, penectomy for pain control, and spinal laminectomy against spinal cord compression due to bone metastases.

### 2.2. Immunohistochemistry

Immunohistochemistry was carried out using anti-AKR1C3 antibody (at a dilution of 1:200; Abcam (ab49680, Abcam plc, Cambridge, UK)). As a positive control of anti-AKR1C3 antibody, we used surgical specimens of breast cancer (estrogen receptor (+) and progesterone receptor (+)) [24]. Immunohistostainings were performed using Ventana Discovery Ultra system (Roche diagnostics) as an automatic immunohistostaining apparatus. All specimens were evaluated by two urological pathologists (S.S. and T.Y.). AKR1C3 immunostaining of benign epithelium was relatively homogenous, whereas that of cancer epithelium was heterogeneous. Thus, the strongest immunostaining intensity of AKR1C3 was compared between benign epithelium and cancer cells at each spot. In order to compare AKR1C3 immunostaining in consecutive specimens of cancer cells in each individual, and to evaluate the correlation of AKR1C3 immunostaining of cancer cells with PSA progression-free survival (PFS) after RP, the pathologists evaluated each of the staining proportion and intensity, and the sum of these evaluation scores was considered as the total score (TS). “Proportion score (PS)” was evaluated according to the expression rate of stained tumor cells as: <1% (score 0), 1%–10% (score 1), 11%–33% (score 2), 34%–66% (score 3), and >67% (score 4). “Intensity score (IS)” was evaluated as none (score 0), weak (score 1), intermediate (score 2), and strong (score 3) in most immunostained cells. The Gleason score (GS) of hematoxylin and eosin staining was also evaluated by the urological pathologists.

### 2.3. Statistical Analysis

Results were analyzed with JMP13 software (SAS Institute Inc., Cary, NC, USA); *p*-values were calculated with the Kruskal–Wallis test, Pearson’s chi-squared test, and Wilcoxon signed-rank test. PSA PFS was estimated by Kaplan–Meier analysis and groups compared with the log-rank test. Cox proportional hazard analysis was used to examine the factors associated with PSA PFS. A *p* value less than 0.05 was considered to be statistically significant.

## 3. Results

### 3.1. AKR1C3 Immunostaining of Cancer Epithelium Is Significantly Stronger than That of Benign Epithelia in Patients with Localized Hormone-Naïve Prostate Cancer

Clinical and pathological features are demonstrated and the results of statistical analysis of correlation between demographic features and AKR1C3 expression are presented as *p*-values in Table 1. Representative immunostaining of AKR1C3 is presented in Figure 1 (Appendix A, Appendix A). The distribution of AKR1C3 immunostaining scores are presented in Figure 2. AKR1C3 immunostaining was significantly stronger in cancer epithelia than in benign ones within the same spots (*p* < 0.0001). No correlation was observed between GS and AKR1C3 immunostaining in each spot (Table 1). These results suggested that AKR1C3 might play a role in PCa occurrence.

### 3.2. AKR1C3 Immunostaining of Cancer Cells Is Statistically Associated with PSA Progression-Free Survival after Radical Prostatectomy

The distribution of TS of AKR1C3 immunostaining in cancer cells from RP specimens is listed in Table 2. RP cases were dichotomized according to the median TS of the AKR1C3 immunostainings as: AKR1C3 TS ≤ 2 and AKR1C3 TS ≥ 3. AKR1C3 immunostainings and PSA PFS after RP were statistically correlated, and cases with a high AKR1C3 immunostaining TS had lower PSA PFS than those with a low AKR1C3 immunostaining TS (*p* = 0.042) (Figure 3). In order to evaluate prognostic factors for PSA PFS after RP, cox proportional hazards regression analysis was conducted with PSA at diagnosis, Gleason grade group, and AKR1C3 expression. AKR1C3 expression was an independent risk factor of PSA failure among our cohorts (*p* = 0.032, hazard ratio = 2.19) (Table 3). These results showed that AKR1C3 expression of cancer cells may be a prognostic marker of patients who received RP.

### 3.3. AKR1C3 Immunostaining Is Significantly Stronger in CRPC Tissues rather than in Hormone-Naïve Ones in the Same Cases

We obtained HNPC and CRPC tissues from the same patients in 11 cases; clinical and pathological characteristics are presented in Table 4. CRPC tissues revealed significantly stronger AKR1C3 immunostaining than hormone-naïve tissues in the same cases (*p* = 0.0234, Wilcoxon signed-rank test; Table 4). Interestingly, the longitudinal specimens at hormone-naïve, hormone-sensitive, and castration-resistant states were evaluated in one patient. Immunostainings of AKR1C3 are presented in Figure 4; AKR1C3 was gradually up-regulated with disease progression. These results implied that up-regulation of AKR1C3 might be required for the progression to CRPC in some cases, and it could be a therapeutic target for this complicated disease.

## 4. Discussion

Based on our immunohistochemical analysis of human prostate tissues, we confirmed that AKR1C3 might be crucial in PCa occurrence and progression. In particular, this is the first study to report that AKR1C3 immunostaining increases along the treatment course, that is, AKR1C3 expression elevates from the hormone-naïve status to the CRPC stage in the same patient. Moreover, our study is unique in showing that the PSA PFS rate of patients with high AKR1C3 expression in cancer cells derived from RP specimens was lower than that with low AKR1C3 expression. The majority of our CRPC cases received LHRH analog collectively with bicalutamide as a primary hormonal therapy. Nevertheless, we could not evaluate the correlation of pharmacological treatment with AKR1C3 immunostaining, since CRPC specimens were mostly obtained just during the time of transition to CRPC, before administration of docetaxel or androgen receptor-axis-targeted agents (ARATs), including enzalutamide and abiraterone. To the best of our knowledge, there is no report of AKR1C3 expression after ARAT treatment; however, AKR1C3 activation, both in vitro and in vivo, using PCa cell lines, has been shown as a factor of resistance against ARATs [25,26].

Lin et al. was the first to report a high-titer isoform-specific monoclonal antibody for AKR1C3 and demonstrated AKR1C3 expression in stromal cells, though only faintly in epithelial cells in normal prostate; however, in PCa cells, elevated expression was observed by immunostaining [23]. The same group also reported AKR1C3 to be positive in immunostaining, in 9 out of 11 PCa cases, and showed variation from strong to negative immunostaining within the same tumors, as in our study [26]. They also found no correlation in staining patterns between AR and AKR1C3 expression, consistent with our study (data not shown) [27]. Tian et al. examined the primary PCa biopsy specimens and showed that AKR1C3 expression by immunostaining gradually increases with an elevated GS in PCa epithelium [16]. In our cases, there was no correlation between GS and AKR1C3 immunostaining, which is incompatible with Tian’s results. This might be because our cases underwent RP and most of them had GSs of less than 8, while in Tian’s report, half of the PCa cases had GSs of 8 or higher. Stanbrough et al. revealed that AKR1C3 expression, as per immunohistochemistry, showed negative-to-heterogeneously weak staining in most primary PCa, but intermediate-to-strong AKR1C3 staining in CRPC specimens, which is in agreement with our results [22]. AKR1C3 expression analysis of cancer cells derived from RP showed immunostaining to be correlated with PSA PFS after RP. The result is reproducible even if we adopt IS of AKR1C3 as a representative of its expression (data not shown). In our knowledge, this is the first report of correlation of AKR1C3 expression with PSA PFS after RP. Additionally, AKR1C3 expression was an independent factor of poor PSA PFS when we analyzed the prognostic factors by multivariate analysis including the initial PSA level and Gleason grade group, which were previously considered to be significant predictors of recurrence-free survival after RP [28]. In future, we should analyze AKR1C3 expression and survival after RP in a much larger cohort to understand its role in clinical practice.

AKR1C3 is a multifunctional steroid-metabolizing enzyme that catalyzes androgen, estrogen, progesterone, and PG metabolism [29]. It reduces DHT to form 3α-diol, which is a neurosteroid that acts as a positive allosteric modulator of the gamma-aminobutyric acid type A receptor (GABA_A_R) [30]. The 3α-diol stimulates AR-negative PCa cells through GABA_A_R. Further, it up-regulates the epidermal growth factor (EGF) family members in AR-negative PCa cells and stimulates EGF receptor and Src. These results together suggest that AKR1C3 modulates intraprostatic neurosteroid that, in turn, activates AR-negative PCa progression. AKR1C3 is known to regulate its expression by ERG via direct binding to *AKR1C3* gene [31]. Furthermore, ERG and AKR1C3 expression in human metastatic PCa tissues was revealed to positively correlate with each other by immunohistochemistry. AKR1C3 is known to regulate the stability of ubiquitin ligase Siah2, and thus enhance the Siah2-dependent regulation of AR activity via non-catalytic function [21]. Wang et al. reported that AKR1C3 could drive epithelial–mesenchymal transition (EMT) by activating the ERK signaling pathways and up-regulating transcription factors such as ZEB1, TWIST1, and SLUG, thereby facilitating PCa metastasis [19]. Therefore, AKR1C3 might be crucial in PCa progression.

There are several limitations in our study. The samples analyzed were relatively small, in particular, in consecutive PCa with CRPC progression. We were unaware of the role of AKR1C3 overexpression, in particular, in progression to CRPC, since we did not analyze the tissue concentrations of T and other androgens. Moreover, we did not know how AKR1C3 overexpression correlated in response to new ARATs, namely enzalutamide and abiraterone. Only three cases received ARATs before CRPC tissue extraction. Case 5 received abiraterone 3 months before laminectomy, case 7 was administered enzalutamide 8 months before TUR-P, and case 11 acquired enzalutamide 10 days before TUR-P.

In conclusion, expression of multifunctional AKR1C3, which is known to be the most up-regulated steroidogenic enzyme in patients with CRPC, might increase with the occurrence and longitudinal progression of the tumor in certain cases [17,32]. Targeting AKR1C3 might overcome the complicated disease, CRPC, and also control resistance against ARATs. We plan to further reveal the functions of AKR1C3 in PCa progression and discover potent and specific drugs to inhibit AKR1C3 function.

## Figures and Tables

**Figure 1 jcm-08-00601-f001:**
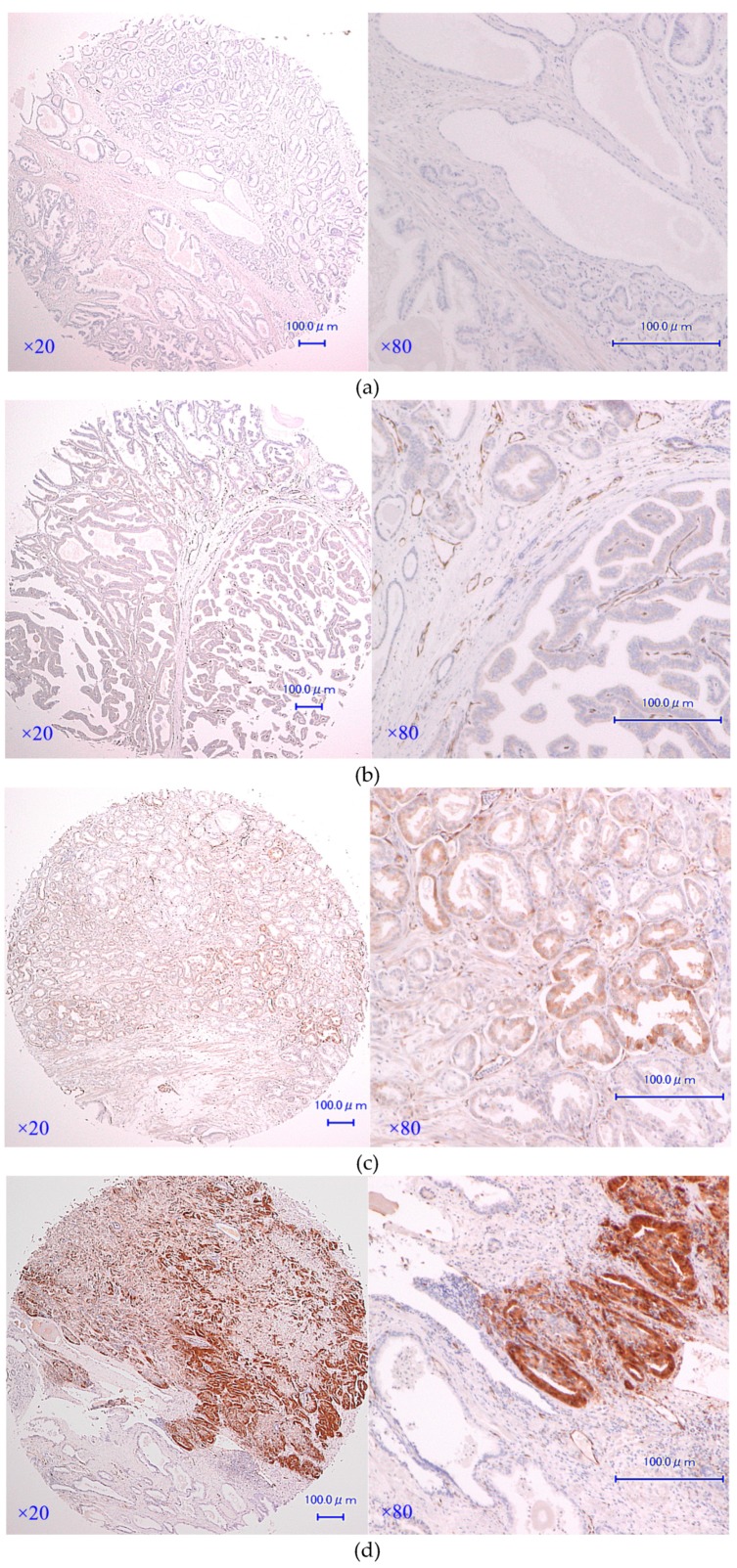
Representative immunostainings of aldo-keto reductase family 1 member C3 (AKR1C3): (**a**) score 0 (none staining), (**b**) score 1 (weak staining), (**c**) score 2 (intermediate staining), and (**d**) score 3 (strong staining).

**Figure 2 jcm-08-00601-f002:**
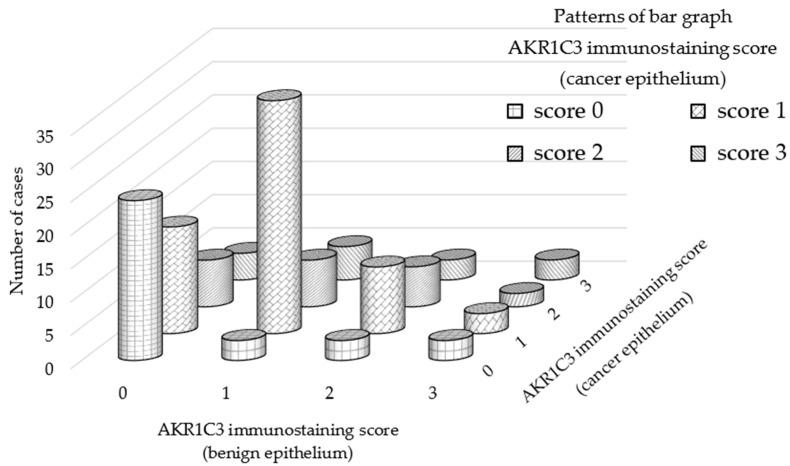
Difference of AKR1C3 immunostaining score between benign and cancer epithelia in the same individuals. AKR1C3 immunostaining was significantly stronger in the cancer epithelia than in the benign ones at the same spots (*p* < 0.0001, Pearson’s chi-squared test).

**Figure 3 jcm-08-00601-f003:**
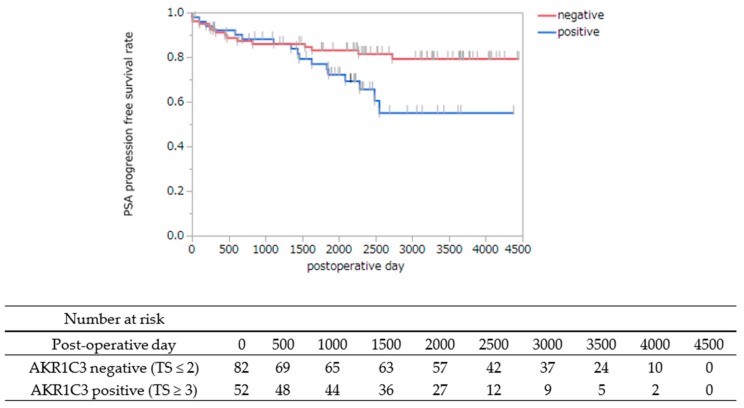
Kaplan–Meier survival curves revealed that the AKR1C3 positive group (TS ≥ 3) had a significantly lower PSA PFS rate than the negative group (TS ≤ 2) (*p =* 0.042, log-rank test).

**Figure 4 jcm-08-00601-f004:**
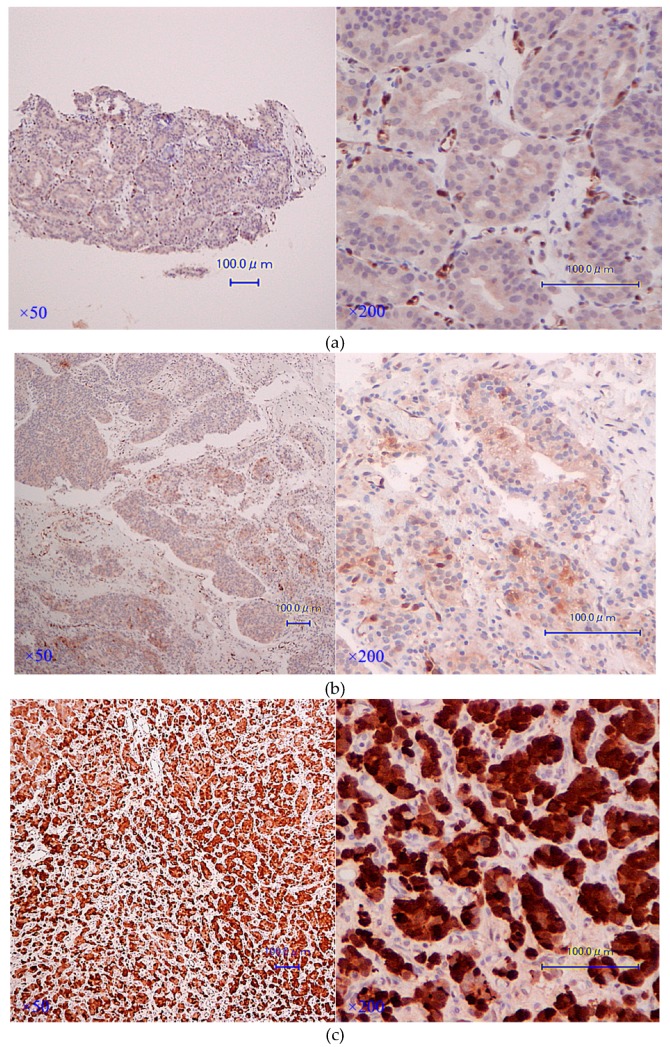
(**a**–**c**) AKR1C3 immunostaining of HNPC, Hormone-sensitive prostate cancer (HSPC), and CRPC specimens in the same case (case 9). CAB (leuprolide acetate + bicalutamide) was initiated for case 9 after diagnosis (Figure 3a): (**a**) HNPC (biopsy) specimens at diagnosis, PSA: 29.9 ng/mL (normal reference range 0-4.0 ng/mL), AKR1C3: Proportion score (PS), 1; Intensity score (IS), 1; Total score (TS), 2; (**b**) HSPC specimens on day 45 after commencing CAB, PSA: 2.89 ng/mL, AKR1C3: PS, 2; IS, 2; TS, 4; (**c**) CRPC specimens, PSA: 46.74 ng/mL, AKR1C3: PS, 4; IS, 3; TS 7. After CAB initiation, transurethral lithotomy (TUL) and TUR-P were performed due to repeated urinary retention resulting from bladder stone (Figure 4b). After 1.5 years of bicalutamide, 2 months of flutamide, and 3 months of ethinylestradiol, together with continuous luteinizing hormone-releasing hormone (LHRH) agonist administration, TUR-P was performed due to urinary retention caused by enlargement of the local tumor (Figure 4c). The immunostaining results suggested increased expression of AKR1C3 in PCa tissues with disease progression.

**Table 1 jcm-08-00601-t001:** Clinicopathological features of tissue-microarray (TMA) specimens with both benign epithelium and cancer cells in the same spot.

*n* = 134		*n*	*p* Value
Age (mean ± SD)		65.6 ± 6.31	0.6087 ^†^
PSA, ng/mL (median)		7.25 (IQR 5.40–9.88)	0.9429 ^†^
Pathological T stage	T2a	7	N.A.
	T2b	1	
	T2c	77	
	T3a	37	
	T3b	12	
Grade group (pathological)	1	47	0.4119 ^††^
	2	38	
	3	37	
	4	9	
	5	3	

*p*-values indicate correlation of expression intensity with AKR1C3 total score; ^†^ Kruskal–Wallis test, ^††^ Pearson’s chi-squared test.

**Table 2 jcm-08-00601-t002:** AKR1C3) total score distribution of TMA specimens.

AKR1C3 Immunostaining	Score	*n* = 134
AKR1C3 total score	0	45
	2	37
	3	11
	4	19
	5	14
	6	4
	7	4
AKR1C3 total score (median)	2 (IQR 0–4)	

**Table 3 jcm-08-00601-t003:** Cox proportional hazard regression analysis of prostate-specific antigen progression free survival (PSA PFS) and clinical and pathological variables.

Variables	PSA PFS Rate		
	HR	95% CI	*p* Value ^†††^
PSA level before RP	1.12	1.05–1.18	0.0003
Grade group	1.66	1.16–2.36	0.0053
AKR1C3 (TS)	2.19	1.07–4.55	0.032

HR: hazard ratio, CI: confidence interval, and ^†††^ Wald test.

**Table 4 jcm-08-00601-t004:** Clinicopathological features and results of AKR1C3 immunostaining total score in 11 cases, including both hormone-naïve and castration-resistant specimens.

Case	Age at Diagnosis	Clinical Stage at Diagnosis	Gleason Score at Diagnosis	Excised CRPC Organ	Age at Excision	PSA at Excision	Days from Diagnosis to Castration	Treatment until Excision of CRPC Specimens	AKR1C3 Immunostaining Total Score	
									at HNPC	at CRPC
Case 1	57	cT3bN1M0	3 + 3	Prostate	62	25	786	CAB + DOC	6	7
Case 2	73	cT3bN0M0	3 + 4	Prostate	84	5	2860	CAB	4	7
Case 3	60	cT3aN0M1c	4 + 4	Prostate	65	392	603	CAB + DOC	4	4
Case 4	79	cT3aN0M0	4 + 4	Penis	82	16.1	1051	CAB	2	4
Case 5	68	cT3aN0M1c	4 + 4	Thoracic vertebra	75	39.21	702	CAB + DOC + Abi	0	5
Case 6	70	cT3aN0M0	4 + 3	Prostate	78	408.2	2112	CAB	6	6
Case 7	76	cT4N1M1b	4 + 5	Prostate	85	14.07	2320	CAB + Enz	4	7
Case 8	78	cT4N1M1b	5 + 4	Thoracic vertebra	79	7.58	321	CAB	7	7
Case 9	69	cT4N0M0	4 + 5	Prostate	71	46.74	567	CAB	2	7
Case 10	80	cT3bN1M0	4 + 5	Prostate	82	6.21	544	CAB	1	1
Case 11	69	cT4N0M1b	4 + 5	Prostate	70	127	318	CAB + Enz	6	5

CAB: combined androgen blockade, DOC: docetaxel, Abi: abiraterone, and Enz: enzalutamide.

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
