# Peer review of "Consecutive Prostate Cancer Specimens Revealed Increased Aldo–Keto Reductase Family 1 Member C3 Expression with Progression to Castration-Resistant Prostate Cancer"

_jcm, 2019, doi:10.3390/jcm8050601_

Reviewer 1 Report

The manuscript by Miyazaki et al. describes the impact of AKR1C3 in prostate cancer progression employing an immunohistochemical analysis of tissue micro arrays. The authors compare the expression levels of AKR1C3 between benign prostatic epithelium and cancer cells and the levels of AKR1C3 between hormone naïve prostate cancer and advanced CRPC as well.

 The methods and the introduction are well described, but, unfortunately, the style of the manuscript is hard to follow and there are several shortcomings in the data presentation. The “Results” are too descriptive, relatively short and does not interpret data sufficiently. In my opinion the authors should complete the results with a brief discussion/conclusion (1-2 lines). The quality of figures 1 and 3 is not good. These images are blurred, and the scale bars are cut.  I am sure that the authors can improve it. Also, authors should show more immunostainings representative of AKR1C3 (for instance putting more images in supplementary section).

Grammar also needs moderate improvement.

Resuming, although the authors draw the right conclusions there is work to do on the manuscript

 Minor points and typos:

- Line 12: … (AKR1C3) acts as a key….

- Line 12: please use the abbreviation PCa for “Prostate Cancer”

- Line 13: I can not understand this sentence: “however, several reports have emphasized on prostate cancer tissues derived from different patients or cell lines”

- Line 32: Testosterone (T)

- Line 36: The sentence which start with “Several possibilities……” is quite redundant and hard to follow. It would be appropriate to re-formulate this sentence.

- Line 45: ….CRPC and has been

- Line 97: immunostaining scores are

- Discussion: I would remove the last sentence

Author Response

The manuscript by Miyazaki et al. describes the impact of AKR1C3 in prostate cancer progression employing an immunohistochemical analysis of tissue micro arrays. The authors compare the expression levels of AKR1C3 between benign prostatic epithelium and cancer cells and the levels of AKR1C3 between hormone naïve prostate cancer and advanced CRPC as well.

The methods and the introduction are well described, but, unfortunately, the style of the manuscript is hard to follow and there are several shortcomings in the data presentation. The “Results” are too descriptive, relatively short and does not interpret data sufficiently. In my opinion the authors should complete the results with a brief discussion/conclusion (1-2 lines). The quality of figures 1 and 3 is not good. These images are blurred, and the scale bars are cut. I am sure that the authors can improve it. Also, authors should show more immunostainings representative of AKR1C3 (for instance putting more images in supplementary section).

Thank you for your kind suggestions. We added some brief conclusion in each result session in line 126 on page 3, in line 156-7 on page 6, and in line 173-175 on page 7. Furthermore, we have improved immunohistochemical figures and additional representative cases in supplementary section.

Grammar also needs moderate improvement.

Resuming, although the authors draw the right conclusions there is work to do on the manuscript

Our manuscript has been checked and edited by a native speaker in Editage (www.editage.jp)

Minor points and typos:

Thank you for your kind suggestions. We have revised all the issues accordingly as below.

Line 12: … (AKR1C3) acts as a key….

We have changed the description to “(AKR1C3) is an enzyme in the steroidogenesis pathway, especially in formation of testosterone and dihydrotestosterone, and is believed to have a key role in promoting prostate cancer (PCa) progression”

Line 12: please use the abbreviation PCa for “Prostate Cancer”

We have used the abbreviation PCa for “Prostate Cancer”.

Line 13: I cannot understand this sentence: “however, several reports have emphasized on prostate cancer tissues derived from different patients or cell lines”

We have omitted the description for clarification.

Line 32: Testosterone (T)

We have used the abbreviation T for “testosterone”.

Line 36: The sentence which start with “Several possibilities……” is quite redundant and hard to follow. It would be appropriate to re-formulate this sentence.

We have omitted the description for clarification.

Line 45: …CRPC and has been

We have changed the description according to the reviewer’s suggestion.

Line 97: immunostaining scores are

We have changed the description according to the reviewer’s suggestion.

Discussion: I would remove the last sentence

We have omitted the description as the reviewer’s suggestion.

Reviewer 2 Report

In the present manuscript, authors compared the expression level of Aldo –Keto Reductase Family 1 Member C3 (AKR1C3), a key molecule for prostate cancer progression, in particular, in castration-resistant prostate cancer (CRPC), between benign prostatic epithelium and cancer cells in the same patients.

In the title " as a Transition" is not appealing and doesn't explain the results obtained.

Introduction section should be improved with a deeper and comprehensive  description of  androgen and estrogen receptors that exert a pivotal role in PC initiation and progression. AKR1C3 induce the production of 3α-diol that can binfd to estradiol receptors. To obtain useful information, authors could read the following manuscripts  doi: 10.3389/fonc.2018.00002, doi: 10.18632/oncotarget.6220. Additionally AKR1C3 promotes epithelial-mesenchymal transition, an important process responsible of metastasis that is often related to CRPC. Please read the following manuscript  10.3389/fphar.2019.00028.

 hormone naïve prostate cancer is unusual. What is the definition of HNPC?
In the table 1, please specify the meaning of pT.
In the material and methods section authors reported "As a positive control of anti-AKR1C3 antibody, we used surgical specimens of breast cancer (estrogen receptor (+) and progesterone receptor (+)". Why the authors decided to use ER positive and PR positive breast sample? As a consequence, the prostate cancer samples analyzed  show a PR or ER expression?
Additionally,  what is the Androgen receptor expression level in the prostate samples analyzed ?
Please add a list of abbreviations

Please check English form

Author Response

In the present manuscript, authors compared the expression level of Aldo –Keto Reductase Family 1 Member C3 (AKR1C3), a key molecule for prostate cancer progression, in particular, in castration-resistant prostate cancer (CRPC), between benign prostatic epithelium and cancer cells in the same patients.

In the title " as a Transition" is not appealing and doesn't explain the results obtained.

We appreciate your thoughtful comments. We have revised the title as “Consecutive prostate cancer specimens revealed increased in Aldo-Keto Reductase Family 1 Member C3 expression with progression to castration-resistant prostate cancer”.

Introduction section should be improved with a deeper and comprehensive description of androgen and estrogen receptors that exert a pivotal role in PC initiation and progression. AKR1C3 induce the production of 3α-diol that can bind to estradiol receptors. To obtain useful information, authors could read the following manuscripts doi: 10.3389/fonc.2018.00002, doi: 10.18632/oncotarget.6220. Additionally, AKR1C3 promotes epithelial-mesenchymal transition, an important process responsible of metastasis that is often related to CRPC. Please read the following manuscript 10.3389/fphar.2019.00028.

Thank you for your suggestions. We have added some descriptions and references in line 55-67 on page 2.

Hormone naïve prostate cancer is unusual. What is the definition of HNPC?

We sometimes use “hormone naïve prostate cancer” as a representative of prostate cancer which is not exposed to any hormonal therapy, as you see in “Hao et al. 2018 Eur Urol PMID 29544736” and “Wallis CJD et al. 018 Eur Urol PMID 29037513”.

In the table 1, please specify the meaning of pT.

Thank you for your kind suggestion. We changed it as “pathological T”.

In the material and methods section authors reported "As a positive control of anti-AKR1C3 antibody, we used surgical specimens of breast cancer (estrogen receptor (+) and progesterone receptor (+)". Why the authors decided to use ER positive and PR positive breast sample? As a consequence, the prostate cancer samples analyzed show a PR or ER expression? 

According to Lin et al. 2004 Steroids, we use ER and PR positive breast cancer tissue as a positive staining for AKR1C3 expression. This does not mean that we will like to evaluate ER and PR expression in PCa and actually we did not check ER and PR expression in our cohort. As your comments we added the reference in “Material and Methods” in line 96 on page 3 as reference #24.

Additionally, what is the Androgen receptor expression level in the prostate samples analyzed ?

We also evaluated AR expression in PCa of our cohort using TMA but there was no strong correlation with AR and AKR1C3 expression. Since evaluation of AR expression is not our aim of this study we did not included the results in our manuscript. We added the description in line 35-36 on page 10.

Please add a list of abbreviations

Thank you for your suggestion. We added the abbreviations in the last part of our manuscript.

Please check English form

Our manuscript has been checked and edited by a native speaker in Editage (www.editage.jp)

Reviewer 3 Report

The goal of this study was to determine whether AKR1C3 expression levels are associated with prostate cancer and the development of CRPC. A key strength is that the study uses matched tumor tissue to assess the association with development of CRPC; AKR1C3 expression levels are compared in needle biopsy tissue taken prior to and after the development of CRPC. While only 11 patients were assessed for this part of the study, the fact that this type of matched analysis has not been performed before makes the finding of interest. There are multiple grammatical errors throughout the manuscript – these are distracting and need to be corrected. Other concerns are listed below;

1.       Line 12: I suggest rephrasing this sentence. Perhaps replace ‘acts as a key molecule for’ with ‘plays a key role in promoting’. The authors may also want to consider including a sentence describing the function of AKR1C3 in the Abstract to help orientate readers.

2.       Line 20: should say ‘from’, not ‘by’

3.       Line 31: This sentence is misleading and needs to be modified – ADT is not used to treat all prostate cancer patients, only those with a biochemical recurrence.

4.       Introduction: This journal has a broad audience and as such the terms HNPC and CRPC should be described in more detail

5.       Line 46: The author should include specific details regarding the reaction(s) that AKR1C3 catalyzes (this information is included in the Discussion – I think it should be moved to the Introduction section to provide the reader with context as to why assessment of this molecule is important).

6.       Section 2.1: More details regarding the collection of tumor specimens needs to be included. I assume that the CRPC specimens were collected from bone metastases prior to treatment for CRPC was initiated - please clarify this in the methods and add more details about specimen collection.

7.       Table 1: the title of this table should be modified to make it clear that this table is comparing AKR1C3 expression in matched normal and HNPC patients. Section 2.2: Please state how many cores were assessed for each HNPC and CRPC specimen - I assume at least three. Patient race should also be noted.

8.       Figure 2: the scale bars for the X40 magnification pictures have been cut off. Please fix this.

9.       Line 98: the phrase ‘at the same spots’ needs to be explained

10.   Discussion section, first line: I think this statement is inappropriate – the authors could perhaps state that their data confirm/align with other studies which have shown that increased AKR1C3 expression is associated with development of CRPC.

11.   Discussion: more time should be spent on comparing the study data with data from published studies.

12.   Discussion: please provide more details describing how ARATs could have impacted AKR1C3 expression and cite relevant literature

13.   Line 42 of the Discussion: perhaps amend this to say the study supports a role for increased AKR1C3 in the development of CRPC and then state why this important/what next steps will be.  

14.   Consider comparing AKR1C3 expression levels with time to biochemical recurrence – this would help further establish whether a link between this molecule and disease progression exists.

 Author Response

1. Line 12: I suggest rephrasing this sentence. Perhaps replace ‘acts as a key molecule for’ with ‘plays a key role in promoting’. The authors may also want to consider including a sentence describing the function of AKR1C3 in the Abstract to help orientate readers.

Thank you for your thoughtful comments. We change the phrase as the reviewer’s suggestion in line 13-16 on page 1. Additionally, we also added some descriptions in Introduction for the reader to understand AKR1C3 function in line 33-39 on page 1 and in line 40-41 on page2.

2. Line 20: should say ‘from’, not ‘by’

Thank you for your comment. According to the limitation of words used in Abstract, we omitted the sentence.

3. Line 31: This sentence is misleading and needs to be modified – ADT is not used to treat all prostate cancer patients, only those with a biochemical recurrence.

We changed the description to “ADT is required to treat advanced prostate cancer and biochemical recurrent cases after curative radical treatment”.

4. Introduction: This journal has a broad audience and as such the terms HNPC and CRPC should be described in more detail

We added some descriptions in “Introduction” for the reader to understand AKR1C3 function in line 33-39 on page 1 and in line 40-41 on page2.

5. Line 46: The author should include specific details regarding the reaction(s) that AKR1C3 catalyzes (this information is included in the Discussion – I think it should be moved to the Introduction section to provide the reader with context as to why assessment of this molecule is important).

Thank you for your thoughtful suggestion. We added some descriptions in line 56-58 on page 2.

6. Section 2.1: More details regarding the collection of tumor specimens needs to be included. I assume that the CRPC specimens were collected from bone metastases prior to treatment for CRPC was initiated - please clarify this in the methods and add more details about specimen collection.

We have described the origin of CRPC specimens in “Material and Methods”, in line 87-89 on page 2 and in line 90-92 on page 3. And we have described more details in “Result”, as you see in Table 4.

7. Table 1: the title of this table should be modified to make it clear that this table is comparing AKR1C3 expression in matched normal and HNPC patients. Section 2.2: Please state how many cores were assessed for each HNPC and CRPC specimen - I assume at least three. Patient race should also be noted.

Thank you for your kind suggestion. We change the title of Table 1 to “Clinicopathological features of TMA specimens with both benign epithelium and cancer cells in the same spot”. Additionally, we added information about construction of TMA in line 80-83 on page 2. All the CRPC specimens were derived from surgical specimens after local and metastatic progressive lesions. HNPC specimens comprised from needle biopsies or TURP specimens described in “Material and Methods”. We also added information about patient race in “Material and Methods” as the reviewer’s suggestion in line 80 on page 2.

8. Figure 2: the scale bars for the X40 magnification pictures have been cut off. Please fix this.

Thank you for your comments. We added the scale bars correctly in the figures and change pictures from X40 to X80 magnification for clarification.

9. Line 98: the phrase ‘at the same spots’ needs to be explained

Thank you for your comment. We changed the description as “within the same spots” for clarification in line 124 on page 3.

10. Discussion section, first line: I think this statement is inappropriate – the authors could perhaps state that their data confirm/align with other studies which have shown that increased AKR1C3 expression is associated with development of CRPC.

We agree with the thoughtful comment by the reviewer. We change the sentence into “Based on our immunohistochemical analysis of human prostate tissues, we confirmed that AKR1C3 might be crucial in prostate cancer occurrence and progression.” in line 17-18 on page 10.

11. Discussion: more time should be spent on comparing the study data with data from published studies.

According to the reviewer’s kind suggestions we added some description in “Discussion” session comparing the study data with one from previous reports in line 30-44 on page 10.

12. Discussion: please provide more details describing how ARATs could have impacted AKR1C3 expression and cite relevant literature

According to the reviewer’s kind suggestions we added some description in “Discussion” session and added two interesting references (#25,26) in line 28-29 on page 10.

13. Line 42 of the Discussion: perhaps amend this to say the study supports a role for increased AKR1C3 in the development of CRPC and then state why this important/what next steps will be.

We should appreciate for the comments. We added some description in the final part of “Discussion” session in line 77-80 on page 11.

14. Consider comparing AKR1C3 expression levels with time to biochemical recurrence – this would help further establish whether a link between this molecule and disease progression exists.

Thank you for this critical comment. We analyzed AKR1C3 expression and biochemical failure of RP specimens and finally revealed that AKR1C3 expression is an independent prognostic factor of biochemical failure after RP. Moreover, cases with high AKR1C3 expression in RP specimens were poor biochemical free survival. Addition of the results, we believe that our manuscript has substantially improve in its quality. We should appreciate again for this thoughtful suggestion. We added some description in “Material and Methods”, in “Results” and in “Discussion” together with a figure (Fig 3) and a Table (Table 3) for presenting our results.

Round  2

Reviewer 1 Report

The referee is now satisfied with the changes and  considers the paper suitable for the publication

Reviewer 2 Report

Authors improved the manuscript according to my suggestions.